# Reduced body sizes in climate-impacted Borneo moth assemblages are primarily explained by range shifts

Chung-Huey Wu [1], Jeremy D. Holloway[2], Jane K. Hill [3], Chris D. Thomas [3], I-Ching Chen [4]* & Chuan-Kai Ho [1,5]*

Both community composition changes due to species redistribution and within-species size shifts may alter body-size structures under climate warming. Here we assess the relative contribution of these processes in community-level body-size changes in tropical moth assemblages that moved uphill during a period of warming. Based on resurvey data for seven assemblages of geometrid moths (>8000 individuals) on Mt. Kinabalu, Borneo, in 1965 and 2007, we show significant wing-length reduction (mean shrinkage of 1.3% per species). Range shifts explain most size restructuring, due to uphill shifts of relatively small species, especially at high elevations. Overall, mean forewing length shrank by ca. 5%, much of which is accounted for by species range boundary shifts (3.9%), followed by within-boundary distribution changes (0.5%), and within-species size shrinkage (0.6%). We conclude that the effects of range shifting predominate, but considering species physiological responses is also important for understanding community size reorganization under climate warming.

[1] Institute of Ecology and Evolutionary Biology, National Taiwan University, Taipei City, Taiwan. [2] Department of Life Sciences, The Natural History Museum, London SW7 5BD, UK. [3] Department of Biology, University of York, York YO10 5DD, UK. [4] Department of Life Sciences, National Cheng Kung University, Tainan City, Taiwan. [5] Department of Life Science, National Taiwan University, Taipei City, Taiwan. *email: chenic@mail.ncku.edu.tw; ckho@ntu.edu.tw

The climate-driven redistribution of species has resulted in novel biological communities, because the rates at which the ranges of species shift are variable, leading to new species compositions and changed interactions within these reshuffled communities[1–4]. Body-size changes associated with community reshuffling may be particularly important, influencing trophic interactions through changes to predator–prey size ratios[5], as well as altering the distribution and transfer of biomass[6,7], food web stability[5,8], and ecosystem functioning[9]. How range shifts alter the size structure of communities may depend on whether there is a body-size cline along the environmental gradients over which species are shifting, and whether range shifts are size dependent. If Bergmann's rule is operating at interspecific and/or intraspecific levels, then larger species and/or individuals are found towards higher latitudes and/or altitudes[10–15], such that range shifts associated with climatic warming may generally reduce the average size of new communities, as small species/individuals colonize and larger species/individuals disappear[16]. If smaller species with faster population growth[17] or larger species with stronger dispersal capacities[18,19] are more likely to shift their distribution, community size structure will be altered accordingly. However, empirical evidence of how climate-driven range shifts alter body sizes in biological communities is surprisingly lacking[16,20,21], particularly for insects that constitute the majority of terrestrial biodiversity and play key roles in energy flow[22,23].

This dearth of information is surprising, given that intraspecific body-size reduction has been proposed as the third universal response to climate warming[24,25]. For ectothermic organisms in particular, body-size reduction is often associated with faster developmental rates at higher temperatures, in accordance with temperature–size rules[26,27]. Body-size reduction may also be related to higher metabolic rates under warming, if the increasing metabolic cost is not compensated by higher food intake[25]. If developmental size reduction is widely observed, declines in community size can be expected with climate warming. However, empirical evidence for intraspecific size change has been controversial[28], particularly for terrestrial insects[29–35], leaving a clear knowledge gap in relation to the potential ecological consequences at the community level.

It is crucial to integrate both interspecific range shifts and intraspecific body-size changes if we are to understand the process of community size restructuring under rapid climate warming[16,20,36]. Theories and empirical studies on warming-induced size structure changes within communities have been primarily focused on the metabolic demands of species and trophic interactions[9,37,38] without explicitly considering composition changes due to range shifts[39]. To the best of our knowledge, integrating both processes has only been tested by experimental studies in aquatic ecosystems[20] and not yet demonstrated by any long-term field evidence or for terrestrial ecosystems.

Here we analyze a unique dataset of geometrid moths on Mt. Kinabalu (4095 m, Sabah, Malaysia; 6°4′ N,116°33′ E), where climate warming has caused species to move uphill[40] but the body-size consequences of this community restructuring and of intraspecific size changes have not been examined. If size shrinkage applies to tropical insects, then this intraspecific process will reduce overall body size of the assemblage. Positive elevational body-size clines have been reported in neotropical moths, with larger species relatively more frequent at higher altitudes[41]. If this size cline applies to moths on Mt. Kinabalu, we predict that range shifts will also reduce the mean size of new assemblages. We find that over 42 years, moth body size, in terms of forewing length, reduces 1.3% on average for species and ca. 5% for assemblages. Positive body-size clines along the elevational transect are observed at subfamily levels and thus species range shifts contribute to the reduction in assemblage size at higher sites. Overall, the assemblage size shrinkage is driven mainly by species range shifts (3.9%) and to a much lesser degree by within-species size changes (0.6%). Range-shift-induced species reshuffling brings substantial size restructuring and assemblages of low biodiversity are particularly susceptible to impacts of range shifting under climate change.

## Results

**Changes in body-size structures of moth assemblages.** Chen et al.[40] resurveyed the moth transect on Mt. Kinabalu in 2007, 42 years after the original study in 1965[42], using the same field protocols and visiting the same sites along the elevation gradient. As Mt. Kinabalu was established as a Malaysian national park in 1964, the habitat remains largely undisturbed. We were thus able to compare the range shifts[40,43] and body-size changes (this study) under climate warming with limited confounding factors. Over the 42-year study interval, moth species moved uphill by 67 m on average, in response to 0.7 °C warming, reshuffling the species composition of assemblages along the transect[40].

We measured forewing length as our metric of insect size, for 277 species (5536 individuals) and 219 species (3053 individuals) from the 1965 and 2007 surveys, respectively (here we report analyses for 4122 female specimens; see Methods). We examined the size structures of seven assemblages (sites) along the transect and changes between the two surveys (Table 1).

Over the 42-year study period, we find that body sizes decrease for species by 1.3% on average (linear mixed model: mean ± SE shrinkage = 0.25 ± 0.04 mm; $t = -6.37$, $P < 0.001$, $n = 3479$ individuals from 109 species across all sites; Supplementary Fig. 1). To compare our findings with other studies, we transformed the forewing length of each species to dry mass, following Garcia-Barros[44]. This produced a decline of 2.8% on average (mean ± SE shrinkage = −0.88 ± 0.17 mg; $t = -5.034$, $P < 0.001$; based on

**Table 1 Number of moth individuals and species collected or measured at the seven study sites**

| Site | Elevation (m) | Individual collected in 1965 (# of species) | Individual collected in 2007 (# of species) | Percentage measured |
|---|---|---|---|---|
| Park Headquarter (HQ) | 1440 | 1999 (212) | 224 (90) | 74.4% |
| Power Station (PS) | 1885 | 2567 (220) | 1391 (170) | 94.4% |
| Kamborangoh (K) | 2260 | 663 (103) | 582 (103) | 88.0% |
| Radio Sabah (RS) | 2685 | 898 (65) | 939 (68) | 84.3% |
| Paka Cave (PC) | 3085 | 70 (9) | 39 (8) | 99.1% |
| Panar Laban (PL) | 3315 | 81 (7) | 264 (10) | 88.4% |
| Sayat Sayat (SS) | 3675 | 100 (5) | 60 (5) | 87.5% |
| | Total | 6378 (293) | 3499 (235) | 87.0% |

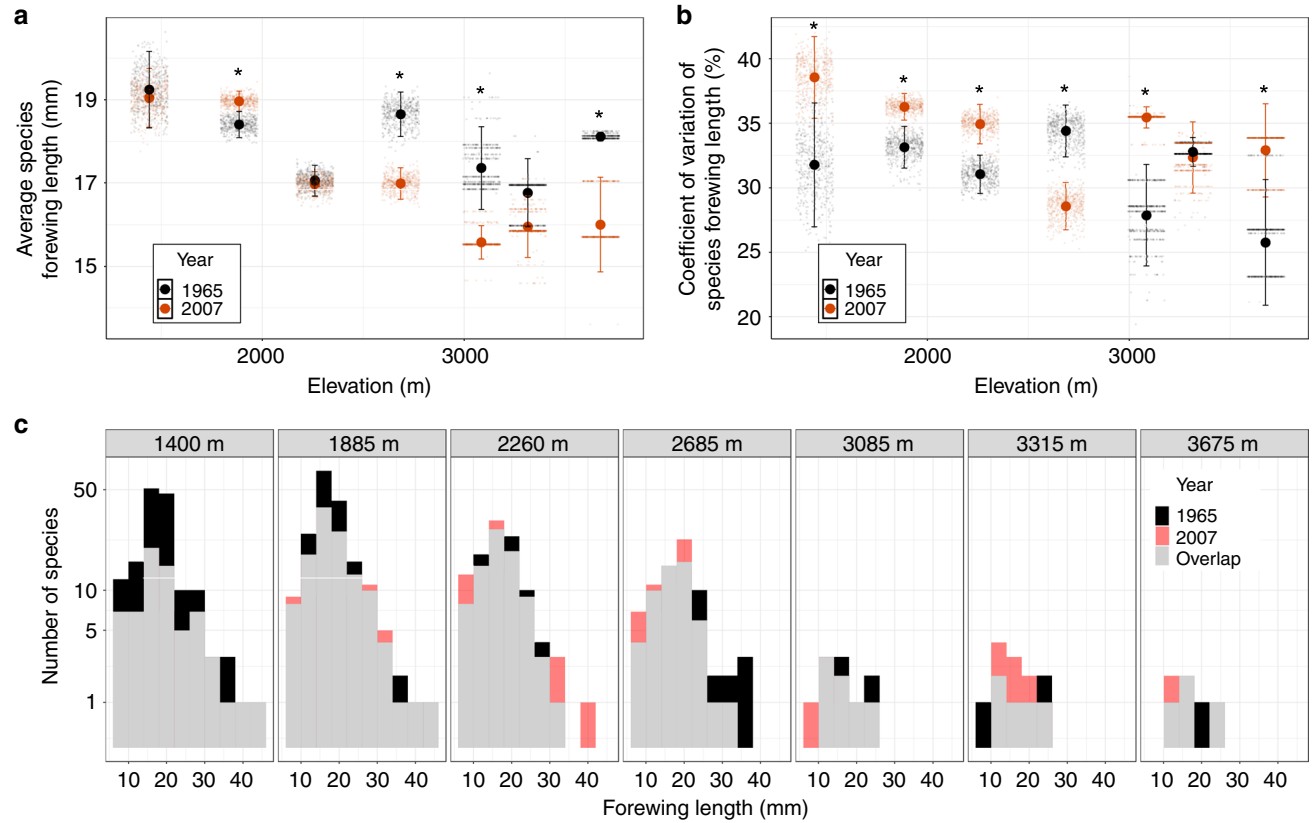

**Fig. 1** Moth assemblage size structure in 1965 (black) and 2007 (red). **a** Average species forewing length (mm). **b** Coefficient of variation of species forewing length. **c** Frequency distribution of species forewing length. In (**a**) and (**b**), mean and 95% confidence interval at each site are shown, based on 500 resamples. Data points are overlaid. Asterisks indicate significant ($p < 0.05$) differences between 1965 and 2007. In (**c**), the number of species are on log10 scale and overlaps between the 2 years are illustrated in gray

0.7 °C warming) or 4% °C$^{-1}$. For assemblages, six of seven showed shrinkage in mean forewing length, three significantly so, although one site (Power Station, 1885 m.a.s.l.) showed a significant increase (Fig. 1a). Changes in the average forewing length at the seven sites varied from $-2.13 \pm 1.10$ mm to $+0.53 \pm 0.42$ mm ($-0.88$ mm on average), representing $-11.7\%$ to $+2.9\%$ changes in body sizes at these sites (Supplementary Table 1). The mean size reduction was most apparent at the highest elevation sites (three sites above 2685 m), where the average body sizes of assemblages were reduced by 0.79 mm to 2.13 mm ($-4.6\%$ to $-11.7\%$). Variation in body size among species within an assemblage is functionally important and so we also considered how the range of sizes present (coefficients of variation of forewing length) at a given site has changed over time. Coefficients of variation increased significantly at five of the seven sites, although one site (at intermediate elevation, Radio Sabah, 2685 m a.s.l.) showed a significant decrease (Fig. 1b, c and Supplementary Table 2). Taken together, these results indicate a trend over the 42-year period for composition shifts towards smaller body sizes at higher elevations, with increased variation in body sizes at most sites.

**Relative contributions of range shift and within-species size change.** The size structures of assemblages are determined by species composition at sites and the body size of each species. Each moth species can potentially alter the composition of new assemblages by range shifts (expansions or contractions at upper or lower boundaries) and non-boundary dynamics (local extinction, local colonization, or local persistence). Thus, each species will contribute to the new assemblage size structure by

range shifts and non-boundary dynamics, as well as by intraspecific size changes under warming (illustrated conceptually in Fig. 2). For each assemblage, to estimate how these processes shape the new size structure, we re-computed the 2007 assemblage size by allowing just one of the following processes to occur at a time (intraspecific size change, four categories of range shift, and non-boundary dynamics). We obtained means and errors for each process's contribution to community size structure change from 500 re-samplings of the dataset to account for differences in sample effort between surveys (see Methods for details).

Overall, we find assemblage size reduction by 4.9%, among which range shifts contribute to an average of 3.9% (ranging from 2.34 mm reduction to 0.34 mm increase, or 12.9% reduction to 1.8% increase across sites), followed by within-species size shrinkage of 0.6% and non-boundary dynamic of 0.5% (Fig. 3a and Supplementary Table 1). Coefficients of variation in body size at sites were also mainly affected by range shifts: range boundary shifts contributed most and increased by an average of 10.7% (ranging from 7.2% decline to 39.8% increase across sites; Fig. 3b and Supplementary Table 2). The reduction of assemblage body sizes at the four sites at highest altitudes was predominantly driven by the upper-boundary expansion of smaller species from lower elevation (Supplementary Fig. 2). By contrast, neither the redistribution of species in-between their upper and lower elevation boundaries (non-boundary dynamics, averaging a reduction of only 0.5% in assemblage body size), nor intraspecific size shrinkage (only 0.6% reduction on average) had important impacts on assemblage body sizes (Fig. 3a and Supplementary Table 1), and had only small effects on body size variation (coefficient of variance (CV); Fig. 3b and Supplementary Table 2).

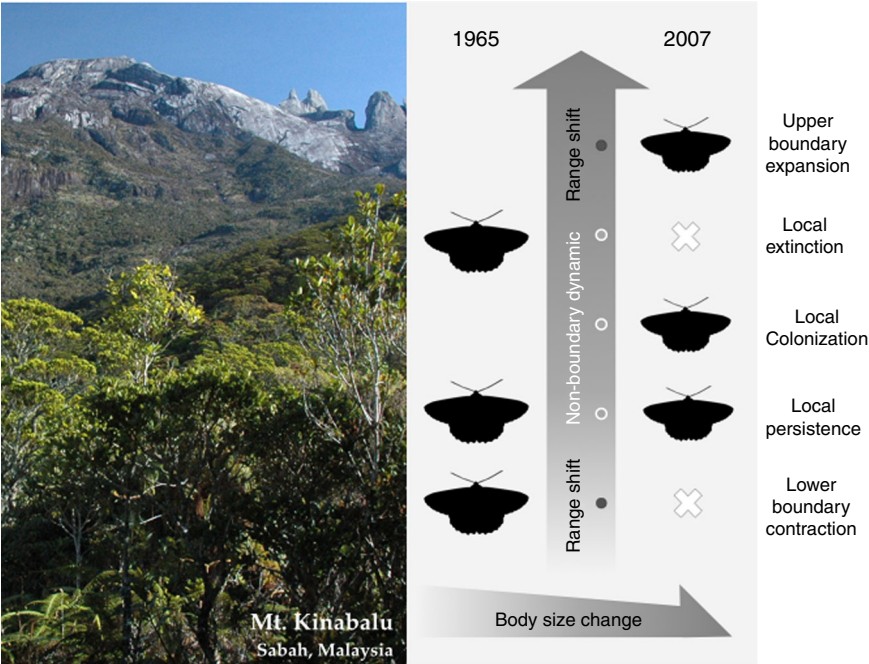

**Fig. 2** Conceptual scheme of how we consider range shifts and body-size changes for each species in this study. Each moth species can potentially alter the composition of new assemblages by range shifts (expansions or contractions at upper or lower boundaries) and non-boundary dynamics (local extinction, colonization, or persistence). The size structure of the new assemblages will be determined by changes in species composition at sites and intraspecific body-size changes of each species. As species are expected to reduce body size and move uphill under warming, assemblage size structures will change accordingly. Photo of Mt. Kinabalu taken by I.-C. Chen, moth silhouette modified from a photo taken by S. Wu

Thus, overall, we found that the reduction in body sizes and increased variation in body sizes in geometrid moth assemblages were primarily driven by the uphill range boundary shifts of small species.

We considered only species presence/absence at sites (assemblages) and not their population abundance at each site, so each species received identical weighting when computing the body-size structure of assemblages. Using the presence/absence data reduces potential biases associated with stochastic population dynamics during the two survey periods and is a conservative approach to the analyses. We repeated these presence/absence analyses by weighting species according to their relative abundances at each site in each year. The results were comparable to those obtained using the presence/absence data, with average sizes of moths in the assemblages shrinking at five out of seven sites (Supplementary Fig. 3a), and coefficient of variation results similar except at the two highest sites (Supplementary Fig. 3b). Dividing the effects into different processes, distribution margin changes still reduced overall assemblage body sizes (a robust result), but there was also a major contribution of non-boundary dynamics (primarily via abundance changes of small vs. large-sized species) in reducing assemblage size profiles (Supplementary Fig. 4a). These tended to increase the within-community coefficient in variation (except for the highest sites; Supplementary Fig. 4b).

## Discussion

Our research provides the long-term field evidence that tropical ectotherms have reduced body size under warming, at least in tropical montane environments. The body size of moths decreased, although it is not yet possible to disentangle how much of the size shrinkage is attributable to phenotypic plasticity, as opposed to evolutionary adaptation[24,25] or to distinguish between direct effects of elevated temperature and indirect effects of potential ecological interactions[25,45,46]. The picture may be complicated by potential interactions between temperature, body size of species, and generation time[47]. We found no correlation between the original size of species and the magnitude of size change (Supplementary Fig. 5), although generation times will likely be shorter at higher ambient temperatures[48] and with warming[49], and feed back into reduced species size. It is also possible that uphill intraspecific gene flow could be contributing to size shrinkage. The magnitude of biomass reduction in our study (2.8%) is higher than the reported reduction of 1.43% °C$^{-1}$ across terrestrial ectotherms[50] or reduction of 0.35% °C$^{-1}$ estimated by laboratory warming[47]. However, taxon-specific dry mass measurements, together with measurements of local temperature changes along the elevation gradient would yield more robust estimates for comparison among studies.

On Mt. Kinabalu, positive body-size clines (with elevation) are observed among species, with clines in the Larentiinae and Geometrinae in 1965, and also within species (Supplementary Figs. 6, 7 and Supplementary Table 3). Uphill range shifts have contributed not only to the decline in mean assemblage size between 1965 and 2007 but have also removed the cline in the Geometrinae (Supplementary Fig. 7). Whether or not range shifts and gene flow reduce the body-size structure of assemblages will depend on the prevalence of among-species and within-species body-size clines along the environmental gradient. Altitudinal size clines among different insect taxa are diverse[13,14], weakly associated with temperature gradients[51], and interact with other confounding factors, including season length, oxygen availability, voltinism, and predation pressure[11,51–55]. As such, a general prediction seems unlikely. In fact, the mechanisms shaping body-size clines may well interact with the tendency of species to carry out range shifts as they respond to climate warming. If the thermal environment acts as the sole or dominant driver of distribution patterns, there may be trade-offs between size change and range shifts[51]. In the case of moths on Mt Kinabalu, the magnitude of range shifts appeared not to compensate

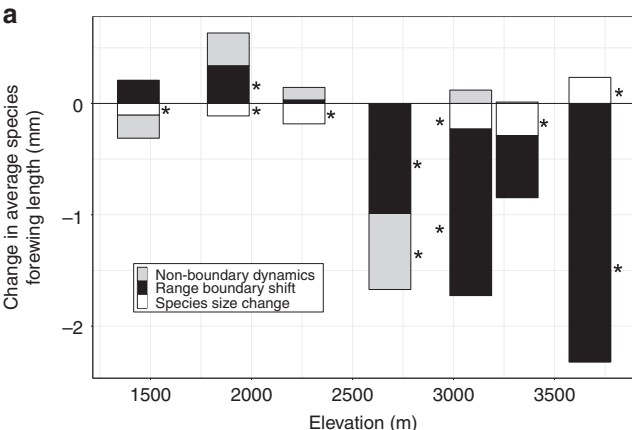

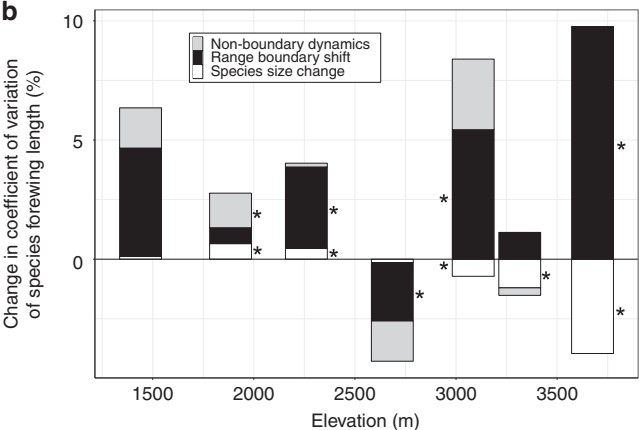

**Fig. 3** Contribution of range boundary shifts (black), intraspecific size change (white), and non-boundary dynamics (gray) to changes in moth assemblage size structure from 1965 to 2007. **a** Change in average species forewing length. **b** Change in coefficient of variation of species forewing length. Asterisks indicate components with effect sizes significantly different from zero at the 95% confidence level, based on 500 resamples

fully for the degree of warming[40]. However, we did not find size-dependent range shifts across all species (Supplementary Fig. 8), nor trade-offs between shrinking body size and changing distribution (Supplementary Fig. 9). The complex interplay of biotic and abiotic factors along the temperature gradients in montane environments may partly explain the idiosyncratic responses of size changes and range shifts among species, despite both being significant across assemblages, and may also explain variation in the responses at different sites.

Not all species contributed equally to the observed changes we report and it is not unusual for widespread species to contribute more to macroecological patterns than narrowly distributed ones[56]. We indeed found that widespread species had greater impact on intraspecific size reductions at the highest elevation sites (Supplementary Fig. 10), primarily because widespread species were relatively frequently observed in these sites (Supplementary Fig. 11). However, although widely distributed species appear to be larger (Supplementary Fig. 12), intraspecific size changes did not correlate with species size (Supplementary Fig. 5). Moreover, the magnitude of intraspecific size changes was small (c.a. ± 0.2 mm) and did not affect our main conclusion that the size structure of new assemblages is primarily driven by range shifts. Most species occupy small ranges and so their impacts on assemblage sizes largely came from boundary shifts, as we found for narrowly distributed species (Supplementary Fig. 13).

Spatial and temporal variations in macroecological phenomena are crucial for understanding how species respond to environmental change. Yet, changes over time have rarely been examined, largely due to the lack of relevant field observations. A possible concern in this study is that we compare two time snapshots and it would be desirable to have multi-year data in future studies. This issue is partly mitigated by the long time period between the two surveys[40,57] but, nevertheless, data for additional years and other research systems are needed to determine the generality of our findings.

Our study reveals that the body-size structures of new assemblages were mainly shaped by species range shifts, which brought smaller species into higher altitude assemblages. Low species diversity at higher elevation sites rendered them particularly susceptible to the impacts of species- and size restructuring under climate change, given that the immigration of a relatively small number of small-bodied species could alter the overall assemblage profile (Fig. 1c). At lower elevation sites with higher biodiversity (and hence more diversity in body sizes among species), the influence of range shifts on mean body size was diluted and more variable. However, variation in body sizes did increase within assemblages at these highly diverse lower elevation sites (1440–2260 m. a.s.l.), potentially altering and broadening the functional structure of these assemblages. The extent to which the biological responses we have observed interact with other factors, such as microhabitat and microclimate shifts in montane environments, and phenology shifts, requires further investigation to establish when different responses exacerbate or compensate for one another[58]. Nonetheless, our results suggest that the redistribution of species and the changing characteristics of individuals are combining to restructure biological communities. The fact that they are doing so in a relatively undisturbed montane tropical forest implies that restructuring could be widespread.

## Methods

**Body-size measurements.** Specimens were retained from the two field surveys (Table 1). Those from the first survey are deposited in Natural History Museum, UK, and those from the second survey are deposited in National Cheng Kung University, Taiwan. For pinned Lepidoptera specimens, forewing length is the best preserved trait that is strongly correlated with overall body size[44] and enables same-stage (i.e., adult) comparisons to be made across years. We measured right forewing length from the wing–thorax junction to the wing tip as an index of individual moth body size. All measurements were performed by the same person (C.-H.W.) using a dissecting microscope with a 0.01 mm precision digital caliper. Specimens with damaged or curled wings or where the sex of the specimen had not been identified were excluded. We were able to collect forewing data from 5536 specimens of 277 species in the historical 1965 survey and 3053 individual specimens of 219 species in the 2007 survey, representing 74.4–99.1% of individuals at a site. Female and male individuals were of similar abundance (female = 4122 individuals and male = 4467 individuals) and were analyzed separately to account for sexual size dimorphism. Both males and females displayed similar body-size change patterns in our preliminary analyses and so we present results for females only, given that functionally important population-level reproductive output and dispersal are more strongly driven by female than by male morphology (number of female individuals = 2928 and 1224 in years 1965 and 2007, respectively). It is noteworthy that the study was exempt from requiring ethical approval for animal testing and research, as it works on preserved specimens of insects.

**Site-specific species forewing lengths.** To account for potential between-population variation in body size across sites (elevation), we calculated site-specific body size as the arithmetic mean of forewing length for each species at a site. These site-specific species forewing length data were used for constructing body-size structure (i.e., species-site data) at the seven sites (see "Shifts in assemblage body-size structure"). Missing data due to damaged or missing specimens were estimated using the same species-site forewing length data in the other survey or, if not available, interpolating the species' forewing length by using the values from the nearest elevation sites in the same year. This two-step estimation increased data availability from 662 to 893 out of 1079 species-site combinations, providing a more complete measure of assemblage body-size structure. Additional analyses showed that the main conclusions of our study are robust under our missing data estimation approach (Supplementary Note 1).

**Shifts in assemblage body-size structure**. To detect a shift in assemblage size structure at a site over 42 years, we examined the site-specific change in average body size (the arithmetic mean of site-specific forewing lengths of species in 2007, minus the same metric for 1965) and in CV of site-specific species forewing length (site-specific CV in 2007 minus CV in 1965). To reduce potential sampling bias due to unequal catch sizes in the two surveys, we applied a re-sampling method to provide robust estimate of average body size and CV at each site in both years. At each site, we randomly sub-sampled the individuals to be 80% of the smaller sample size of the 2 years (modified from Chen et al.[40]). We pooled the results from 500 re-samplings to generate the mean and 95% confidence interval for the following: (1) the average body size and CV of each site in each year and (2) changes in average body size and CV between 1965 and 2007 at each site.

**Intraspecific body-size change**. To examine whether the body sizes of geometrid moths had changed after 42 years of warming, we used raw forewing length data of individual specimens to fit a linear mixed model. The model included a fixed effect (year) and a random intercept effect (elevation-specific population nested within species nested within genus) to account for variations in size and size change, both within- and between-species, and with elevation. We include only species with forewing length data available in both years ($N = 109$ species). Statistical modeling was conducted using package lme4 in R version 3.5[59].

**Relative contributions of range shift and within-species size change**. The observed changes in assemblage size structure (average body size and CV) were partitioned into individual contributions from intraspecific body-size change and range shifts by re-computing assemblage body-size structure under different scenarios. We first defined species composition changes associated with shifts in species' ranges (at their boundaries and sites within the elevation bounds; see below), and quantified how these range-shift-induced composition changes altered average body size and CV at each site (elevation). Then, we compared these range-shift effects with the changes resulting from species body-size change, which were also quantified using the same re-computing approach (see below).

**Range-shift-induced species composition change**. Site-specific colonization and local extinction events, derived from species' presence/absence status at each site in 1965 and 2007, were categorized into range shifts and non-boundary dynamics of each species. Range shifts can be categorized into four types: upper-boundary expansion, upper-boundary contraction, lower-boundary expansion, and lower-boundary contraction. Non-boundary dynamics occurring at sites within each species' elevational limits, including local colonization, local extinction, and local persistent were grouped into one category (Fig. 2). We included catch records from the full transect to obtain correct boundary information. For example, if a species in 2007 colonized a site that was above its elevational range in 1965, this species at this site was classified as belonging to the upper-boundary expansion group at that site. Similarly, if a species went locally extinct from one or more sites at its upper boundary (i.e., recorded in 1965 but not in 2007), it was classified as belonging to the upper-boundary contraction group at this site. Any changes in occurrence at sites of intermediate elevation (in-between the upper and lower bounds) were allocated to non-boundary dynamics at the intermediate-elevation sites. Some species recorded at the lowest four focal sites (HQ, PS, K, RS; 1440–2685 m a.s.l.) in 2007 were previously absent from the study transect (2.9–18.2% of species) and it was difficult to determine whether they were expanding upwards from sites at elevations below the transect or were already-present species that were only dis-covered in the second period. We conservatively grouped changes by such species into non-boundary dynamics, meaning that the "true" total boundary change effect may be slightly higher than our estimate of 3.9%. Such categorizations were per-formed for each of the 500 re-sampling runs, so that potential sampling errors were accounted for.

**Partitioning assemblage body-size structure change**. We partition changes of assemblage body size into six individual components—intraspecific changes of species body size, composition change due to four types of range boundary shifts, and non-boundary dynamics, by re-computing assemblage body-size structure under scenarios in which only one component occurred at a time. For example, to quantify how species' upper-boundary expansion affected assemblage size struc-tures, we first set both species body size and species composition to be identical to those in 1965 and then only let the species in the upper-boundary expansion group colonize sites and alter species composition based on their presence/absence status in 2007. This represents a scenario that only upper range boundary expansion events occurred over the 42 years. The resulting shift in assemblage body-size structure would then be solely caused by species' upper-boundary expansion. Similar calculations were conducted to quantify the contribution of other categories of change. For each component, we calculated changes in average body size and CV (both the absolute value and in percentage) at each site, with the mean and the SE generated from the 500 re-samplings.

**Reporting summary**. Further information on research design is available in the Nature Research Reporting Summary linked to this article.

## Data availability

The raw data are available in Figshare with https://doi.org/10.6084/m9.figshare.9728411. The source data underlying all figures (except conceptual Fig. 2) are provided as a Source Data File.

## Code availability

The R code used in this study is available on request from C.-H. Wu (chung.huey. wu@gmail.com).

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

## Acknowledgements

We thank Y.-T. Lin, C.-H. Hsieh, S.-H. Yen, and S. Wu for constructive comments on this study. We are grateful to G. Martin, J. Chainey, C.-Y. Ho, F.-J. Sha, Y.-C. Chung, and K.-C. Ho for their logistic supports. The photo of Mt. Kinabalu in Fig. 2 is taken by I.-C. Chen; the moth silhouette is modified from a photo taken by S. Wu. I.-C. Chen was sponsored by National Cheng Kung University and Ministry of Science and Technology (MOST) (103-2621-B-006-004 and104-2311-B-006 -006 -MY3). C.-K. Ho was sponsored by National Taiwan University (NTU), College of Life Science, NTU, and MOST (102-2628-B-002-005-MY3 and108-2621-B-002-003-MY3).

## Author contributions

C.-H.W., I.-C.C., and C.-K.H. designed the study. C.-H.W. measured the samples and analyzed the data. J.D.H. provided taxonomic guidance and input to Supplementary Table 3 and Supplementary Note 2. C.-H.W., I.-C.C., and C.-K.H. prepared the manuscript with editing, comments, and interpretation from J. D.H., J.K.H., and C.D.T.

## Competing interests

The authors declare no competing interests.
