## [Peer Review File · Nature Communications]

Reviewers' comments:

Reviewer #1 (Remarks to the Author):

The authors provide a fascinating analysis of size changes over a 40+ year period for Mt Kinabalu moth assemblages. The work is of considerable potential interest because of the combination of intraspecific size change and an analysis of what effectively amounts to differences in beta-diversity among years.

The importance of these contributions to the form of assemblage-level variation in body size has been raised previously in work on assemblage and intraspecific differences in size. It's a surprise to see no reference to either B.A. Hawkins' consideration of these matters (e.g. *Oecologia* 1995; *GEB* 2006) or K.J. Gaston's analysis of these questions (*JB* 2008; *Biol. Rev.* 2010).

A few matters need to be resolved though before the study's conclusions can be considered reliable.

1. The study has effectively two time points. In consequence, no trend can be firmly established since trend data require, at the minimum, a third sampling point (which incidentally reveals why the 2009 PNAS paper is less convincing than might otherwise be the case – 67 m change on average over a ca. 2000 m gradient seems well within stochastic range dynamics). All ecological systems are inherently variable on a year-to-year basis. What cannot be distinguished here is the extent to which the changes detected are representative of interannual variation rather than any trend.

Had the changes been consistent and strong in each of the elevational samples, the outcome may have been considered perhaps more reliable. But as matters stand, of the seven sites, only three showed a significant trend in the direction of a decline in size of the assemblage. One site showed the opposite, and the other three showed no change at all. In the context of two time points, that is not a convincing outcome.

On line 73-74 the authors average out the change and report a mean decline of 0.88 mm (4.9%). What is not clear here is whether species were then counted more than once given that some species occupy more than one elevational band. How much does that actually affect the outcome? Species used in assemblages in this way really cause something of a problem with non-independence as B.A. Hawkins has recently shown (*JB* 2016). The same question comes to the fore in the mean reported on lines 79 to 80.

2. The significance of the change in CV is not actually explained. There's a one line statement (L. 81) with no support. The CV is a measure of the relationship between the mean and standard deviation, so could be influenced by changes across both ends of the distribution. In addition, standard CV calculations are influenced by the extent to which the shape of the distributions changes among years from normal to log normal. It would be useful to better understand why the authors consider the change in CV as evidence for the alteration in size. Again, with two time points one might expect a change.

3. The intraspecific analysis is puzzling since it appears that all individuals are entered into a single model and then an effect of year shown which is presumably a mean of the species effects from the model? This is not clear. In Figure 2 an illustration is made of the individual species effects. Presumably here one could just test the deviation of the slope and intercept from an expected line of identity. That would be simple enough and if there is an expected effect of relatedness then a PGLS-type approach would take care of that. Simply saying that more points lie below than above is not especially convincing. On line 91 arguing that the decline is in line with experimental warming and other field studies is not informative. What does in line mean? Statistically equivalent to size change driven by a similar temperature change as on Mt Kinabalu (which from the 2009

PNAS paper by the authors is estimated at 0.7C). The authors would, incidentally, also be well served by further examination of the terrestrial insect data since several more studies than they cite have shown size changes with temperature and the complexities associated with these.

4. Figure 3 was a little unconvincing. It would be useful to see the range dynamics laid out according to body size class. That would provide a much stronger line of support for the size related range boundary change contention than the current figure. Again, it is not possible to see how much non-independence is introduced by species that occur over multiple sites. Indeed, the concluding sentences about the significance of richness should be considered in light of the role of species that are widespread across the elevational extent compared with those that are not. Without dissecting out this pattern, it is difficult to consider the outcomes reliable.

5. The authors would be well served by considering, again, the established literature on size patterns as opposed to the mechanisms explaining them. Bergmann's rule, for example, has several mechanistic explanations, and for ectotherms these take a variety of forms. Understanding what the case is within the more widely distributed relative to the narrowly distributed species might help to disentangle these effects. The intraspecific analysis could be usefully adapted to do so. That would provide perhaps more insight than the loosely associated discussion of Bergmann's rule, the temperature-size effect and related concepts. The discussion does not really provide clear evidence of the mechanisms the authors have in mind beyond some general notion that temperature affects ectotherm size. Although the material in Table A2.1 and Figure A2.1 (but also remembering that the altitudinal variation figures is an unspecified mix of intraspecific and interspecific information) go some way towards doing so the general discussion really does not provide the kinds of insights the data would allow. Moreover, these data also suggest that partially out the impacts of widespread versus localized species might be useful.

Reviewer #2 (Remarks to the Author):

Wu and co-authors use wing-length data for over four thousand moths collected across an elevational gradient on Mt. Kinabalu, Borneo, in 1965 and 2007 to provide what they claim is the first empirical evidence of within-species reductions in body sizes for tropical insects. The unique dataset allows the authors to separate direct effects of reduced body size from indirect effects on assemblage size structure caused by geographic range shifts. Overall I found the paper to be well written, and mostly clear in its description of the problem and the author's strategy to address the problem. This really does appear to be quite novel use of the data, and I enjoyed reading the manuscript. Below I have listed some general comments I have about the paper, followed by more specific comments that I think could be more easily addressed.

General comments:

My main criticism of the paper is its lack of attention to other possible causes of the observed patterns. I get that the temperature increased on average 0.7C over the past 42 years, but how do you know the changes in moth size were driven by this change? The only time I can see reference to other possible factors is on lines 139-140, where you say there was 'limited habitat change', but if you want to lay out these results as a direct consequence of climate warming then these issues need to be address.

In a similar vein, what is the possibility that your observed results are highly dependent on the specific time interval you analyzed, and if you had analyzed assemblage size structure over a different time interval then could you have observed an opposite effect? Trait distributions in natural communities can respond very quickly to environmental change, and so moth wing lengths may shift on a much smaller time frame than the 42 years you actually observed. The work by Rosemary and Peter Grant on Darwin's finches is an excellent example of how quickly trait distributions in local communities can fluctuate over quite short time frames.

I couldn't find details about how you obtained your climate data – was it predicted from climate models, or based on direct measurement of temperature (i.e., weather station)? Do you have climate data for each site? If so, can you explore temperature changes at each site with their relative change in moth size, to get a more nuanced picture of how temperature relates to changes in size structure?

I think the Abstract needs work. The description of the problem is not as well laid out as in the first paragraph of your Introduction. I think restructure the your Abstract to be more in line with this text.

Have you thought about moving Figure A1.1 to the main text? I think it would clarify quite a few non-intuitive points for the reader (especially the boundary/non-boundary metrics you explore). If you have space in the main text, then you could either move do it in your results section as is (~Line 100), or perhaps better would be to modify the figure to make it a more general introduction to the effects of species redistribution on size structure even earlier when you first talk about these issues (~Line 52-53).

Lastly, perhaps I missed it, but I didn't find anything about how the moths were sampled? I think this is important, just to confirm that sampling was comparable across the two samples.

Specific comments

Line 20: Be specific with what you mean by 'reduce the body sizes of animal species' – do you mean individuals, populations, cohorts, mean community size etc? or all of the above? This becomes clear after reading the paper, but it should be clear from the outset.

Line 25: How many species/taxa?

Line 30: Over what time period was this 4.9%?

Line 48: I'd change 'are likely' to 'appear to be', if that's what previous research shows.

Line 52-53: The second sentence of this paragraph needs to be more clearly linked to the first. Do you mean in general, or for terrestrial systems only?

Line 61: I think you need a dedicated sentence describing how climate of the region has changed between surveys, before you say how much moth assemblages were altered.

Line 84-85: You keep saying the site names, but they aren't shown in your Fig 1 so it seems a little redundant. If they aren't necessary, could just call them 1 though 7, or just by their elevations.

Line 88-91: This is for the entire dataset, but did you find similar results within sites, or you just don't have the power to explore within-site patterns for specific species?

Line 89-90: Its very difficult to see this 1/3% change from the current figure format – is there not a better way to present these data to better highlight the pattern of interest?

Line 114: Climatically warmer?

Line 128-130: So what are the consequences of this for global responses to climate change – e.g., are colder and less diverse high-latitude habitats more likely to have their size structure altered?

Line 136-142: Some of this text seems to be irrelevant to what I was expecting in here – a description of where you got the moth data, and how that data was collected in the field.

Line 179-180: I think mention this in the main text, it's a pretty important caveat to highlight for the reader. I might also add text from Lines 182-184, stating that using pres-abs or abund data didn't change results qualitatively.

Line 212-221: Again, I think some of this could be moved to the main text, perhaps even the figure.

Figure 2: I'm not sure this is the best format to show whether on average species sizes get bigger or smaller – its tough to see whether more points lie above or below the red line.

Figure 3: In your caption I think your reference to Table A1.3 for panel a) should be Table A1.2, and panel b) should be Table A1.3?

Table A1.1 – If you had space, I'd think of moving this table to the main text – its pretty important to the story. Can you also add what the temp changes were at each site (or you only have entire

region?).

Figure A1.1 – This figure is quite relevant – do you have space in the main text for it?

Reviewer #3 (Remarks to the Author):

This study appears to build on an earlier publication by the same research group (Chen et al. 2009. PNAS 106: 1479-1483), which reported long-term (1965-2007) elevation increases in moth assemblages along an altitudinal gradient in Borneo. Here the authors extend their previous work, examining how the mean body size of moth assemblages has changed over time. Importantly, they investigate the relative contribution of range shifts and intra-specific body size reductions to the observed patterns.

At the assemblage level, the authors observe significant reductions in the mean size of moths. These reductions appear to be in response to climate warming (+0.7°C over a 42 year period), primarily driven by range boundary shifts (smaller species moving towards higher colder altitudes), and to a lesser extent body size reductions within species.

I agree that integrating data on species redistribution with physiological responses to temperature (i.e. intra-specific body size reductions) is important for understanding how organisms will respond to climate warming (not forgetting shifts in phenology too). These types of studies are underrepresented in the literature, probably because we lack the necessary long-term (decadal-scale) data to examine such patterns. Consequently, I think this is a valuable study.

That being said, there are several areas in which the manuscript could be strengthened, as well as some caveats that should be better acknowledged. I outline these below.

- The paper would benefit from restructuring. In particular, I feel it is lacking an explicit set of testable hypotheses. Only after their results (line 111) do the authors predict how the direction of body size clines along altitudinal gradients (e.g. Bergmann's rule) will influence shifts in assemblage size structure with warming. Given the predominantly positive altitudinal-size clines on Mt. Kinabalu (i.e. smaller species and/or individuals generally found at warmer, lower elevations), the authors can make some reasonable predictions about how range boundary shifts are likely to contribute to changes in mean assemblage body size. In contrast, where altitudinal-size clines are largely absent, one might predict that range shifts may be less influential in driving changes in assemblage size structure. The authors could do a much better job of setting up these broader hypotheses, before describing how they use moths on Mt. Kinabalu as a case study to test them.
- Long-term patterns in body size can be strongly confounded by variation in life stage and age structure between years. For this reason, it is important to measure the same size-at-stage across years. The life cycle of Lepidoptera make them particularly suited for avoiding these issues, especially when sampling the winged adult form. Nevertheless, this is a very important methodological consideration that should be highlighted to the reader - especially if others intend to investigate these issues in other species using preserved collections.
- It would be useful to comment on the average temperature gradient along the elevation transect. Whilst moth species moved uphill by 67m on average, how does this compare to the gradient in temperature? To what extent do these range shifts compensate for the 0.7°C of warming? I see some of this was briefly discussed towards the end of their earlier publication (Chen et al. 2009. PNAS 106: 1479-1483), but it is particularly important here. Though not the biggest contributor, intra-specific reductions in size were still observed – the persistence of this physiological response would suggest that range shifts do not fully compensate for the increase in temperature. It's interesting that species aren't shifting their distribution to completely offset the physiological effects of warming. Why might this be the case? This should form a more important part of the Discussion.

- Following from the above, did species that exhibited smaller range shifts exhibit stronger intra-specific reductions in body size? Is there a trade-off between these two responses; are they negatively correlated? This type of analysis would be really valuable and particularly novel.
- Figure 2: The authors provide a 1:1 reference line, but I'd also like to see a reduced major axis (RMA) regression through the data (with information about the intercept and slope). This would be particularly interesting given that intra-specific size changes have been shown to vary systematically between species based on key life history traits, including body size. Eyeballing the data, it looks like smaller species might show stronger reductions in size – was this tested? Similarly, do the species in the data set vary in generation time? Previous work on arthropods has shown that smaller multivoltine terrestrial species often reduce their size with warming, whilst larger univoltine terrestrial species generally increase in size (likely driven by a concurrent increase in season length).
- Line 195: Here the random effect is on the intercept only, but not the slope. As I understand it, this accounts for variation in the mean size of species, but assumes that the rate of change (i.e. the slope) between years does not vary between species. Relating to my comment above, did the authors consider investigating how these slopes varied between species? Are there any systematic differences in the magnitude of range shifts and intra-specific size reductions between species? Incorporating these kinds of questions/analyses could really enhance the work.
- It is nice to see that the authors acknowledge and test the robustness of their methodology, in particular with regards to the estimation of missing data (Appendix 3). However, the authors could do much more to acknowledge the limitations of their study in the main text. Whilst the size of the data set is considerable, ultimately the conclusions are derived from 2 points in time (1965 and 2007). This makes it difficult to tease apart the potential drivers, as we cannot tell how closely assemblage size structure has tracked changes in environmental variables over time (namely temperature but also other variables). I appreciate this is the nature of the available data, but this needs to be better acknowledged.
- Be careful when emphasizing novelty (e.g. line 26). Consider rephrasing, as intra-specific body size reductions in tropical insects have been investigated in the lab, across latitude and intra-annually across seasons. Here the novelty seems to come from the investigation of long-term (decadal) changes in size. Be more explicit.

Overall this is an interesting study. I hope my comments are constructive and help to improve the work.

NCOMMS-18-32906-T

Responses to referees' comments

Reviewer #1 (Remarks to the Author):

The authors provide a fascinating analysis of size changes over a 40+ year period for Mt Kinabalu moth assemblages. The work is of considerable potential interest because of the combination of intraspecific size change and an analysis of what effectively amounts to differences in beta-diversity among years.

The importance of these contributions to the form of assemblage-level variation in body size has been raised previously in work on assemblage and intraspecific differences in size. It's a surprise to see no reference to either B.A. Hawkins' consideration of these matters (e.g. *Oecologia* 1995; *GEB* 2006) or K.J. Gaston's analysis of these questions (*JBI* 2008; *Biol. Rev.* 2010).

Our response:

We add further references throughout the main text. We cite Hawkins & Lawton 1995 and Rodríguez et al. 2006 to provide background information about the observation of the Bergmann's rule. We also cite Gaston et al. (2008) and include further analysis to consider how widespread and narrowly-distributed species contribute to intraspecific, interspecific and assemblage level changes (lines 208-218). We cite Chown and Gaston (2010) to explain the diverse altitudinal size cline and confounding factors along the mountain gradient (lines 47, 185-187).

A few matters need to be resolved though before the study's conclusions can be considered reliable.

1. The study has effectively two time points. In consequence, no trend can be firmly established since trend data require, at the minimum, a third sampling point (which incidentally reveals why the 2009 PNAS paper is less convincing than might otherwise be the case – 67 m change on average over a ca. 2000 m gradient seems well within stochastic range dynamics). All ecological systems are inherently variable on a year-to-year basis. What cannot be distinguished here is the extent to which the changes detected are representative of interannual variation rather than any trend.

Our response:

The concern about snapshots of two time points and stochasticity is

understandable. However, we focused on assemblage-level analyses. For inter-annual variability to be a serious concern would require many species to respond in almost the same way to environmental stochasticity, which we know is not true in our study (species vary in the amount they shifted) and in other studies of insects responding to climatic variability (e.g., G. Palmer et al. 2017 *Phil Trans R Soc B* 372, 20160144). The fact that the range shift is robust in the face of a null randomisation approach confirms that the average shift across multiple species is robust, as published in our previous PNAS paper. However, we have added the caveat that data for additional years and other research systems are still needed to determine the generality of our findings. Please see Line 200-206.

Had the changes been consistent and strong in each of the elevational samples, the outcome may have been considered perhaps more reliable. But as matters stand, of the seven sites, only three showed a significant trend in the direction of a decline in size of the assemblage. One site showed the opposite, and the other three showed no change at all. In the context of two time points, that is not a convincing outcome.

Our response:

The trend is the same at 6 of the 7 sites (despite only 3 sites showing individual significance), and the text (including abstract paragraph) is clear that the effect is 'especially at high elevations' (where 3 of the 4 are significant). The prediction (hypothesis) that body size reduction is expected to be most marked at higher elevations is also explained more fully.

The variation is part of the interest. The fact that range shifts are somewhat idiosyncratic and not always size dependent results in diverse patterns of assemblage size restructuring. We now make this clear and discuss expected changes along the gradient according to our hypothesis, as well as confounding factors associated with montane gradients. Please see Lines 122-135, 179-198.

On line 73-74 the authors average out the change and report a mean decline of 0.88 mm (4.9%). What is not clear here is whether species were then counted more than once given that some species occupy more than one elevational band. How much does that actually affect the outcome? Species used in assemblages in this way really cause something of a problem with non-independence as B.A. Hawkins has recently shown (JBI 2016). The same question comes to the fore in the mean reported on lines 79 to 80.

Our response:

Thank you for this comment. We add new analyses to address two aspects of this issue. We include a specific analysis that considers the role of species that vary in range size (please see APPENDIX 4. – WIDESPREAD VS. NARROWLY DISTRIBUTED SPECIES IN ASSEMBLAGE BODY SIZE CHANGE), and add further discussion (in lines 208-218). We also include a new analysis of elevation-specific population, nested within species, nested within genus, to minimize the likelihood of population-level and taxonomic-level pseudoreplication (see our response to point 3).

However, there is no universal solution to this issue because changes to the intraspecific body sizes of widespread species do affect the size distributions of multiple communities. Our new results show that intraspecific size declines (at the assemblage level) were mostly influenced by the widespread species at high elevation sites, given the relatively low species diversity of these sites and that many summit species are widely distributed. By contrast, narrowly distributed species influenced all assemblages along the gradient through range shifts. However, the effect sizes are such that the widespread-species results do not affect our main conclusion that assemblage size changes are primarily driven by range shifts.

2. The significance of the change in CV is not actually explained. There's a one line statement (L. 81) with no support. The CV is a measure of the relationship between the mean and standard deviation, so could be influenced by changes across both ends of the distribution. In addition, standard CV calculations are influenced by the extent to which the shape of the distributions changes among years from normal to log normal. It would be useful to better understand why the authors consider the change in CV as evidence for the alteration in size. Again, with two time points one might expect a change.

Our response:

We consider the CV to be interesting because the range of body sizes present in an assemblage may have as much (or more) functional importance as the average body size. We calculated CV rather than other metrics of variation to provide independence from the average. However, we understand that CV is only one of a number of metrics that we could have used, and hence we have re-analysed the data using SD. The SD generates very similar results to those for CV and thus were not included in the main text. (see below Figure R1). We have not

included this in the paper, but could add it to the supplementary materials if the referees and editors wish.

Figure R1 Contribution of different mechanisms to variation of assemblage size changes calculated by coefficient of variance (upper) and standard deviation (lower).

3. The intraspecific analysis is puzzling since it appears that all individuals are entered into a single model and then an effect of year shown which is presumably a mean of the species effects from the model? This is not clear. In Figure 2 an illustration is made of the individual species effects. Presumably here one could just test the deviation of the slope and intercept from an expected line of identity. That would be simple enough and if there is an expected effect of relatedness then a PGLS-type approach would take care of that. Simply saying that more points lie below than above is not especially convincing. On line 91 arguing that the decline is in line with experimental warming and other field studies is not informative. What does in line mean? Statistically equivalent to size change driven by a similar temperature change as on Mt Kinabalu (which from the 2009 PNAS paper by the authors is estimated at 0.7C). The authors would, incidentally, also be well served by further examination of the terrestrial insect data since several more studies than they cite have shown size changes with temperature and the complexities associated with these.

Our response:

Many thanks for the suggestion. In order to better explore the size change, we tested whether we can improve the linear mixed model to accommodate species specific responses (as also suggested by reviewer #3).

We re-ran the model to include a fixed effect (year) and random intercept and slope effects (elevation-specific population nested within species) to account for variations in size and size change both within and between species, and along the elevation. With this model, the body sizes of individual species decreased by 1.3% on average (mean \pm SE shrinkage = 0.23 ± 0.07 mm; $t = -3.511$, $P < 0.01$, $n = 3479$ individuals from 109 species across all sites over the 42 year period). The year effect did not vary among species (estimated random slope variation = 0.043 ± 0.21). However, a full PGLS-type analysis was not feasible for such an unresolved phylogeny (we tried but the model failed to converge).

Since the species-specific slope effect was not significant and the results are broadly similar, we excluded the random slope effect. In the main text, we reported the model including a fixed effect (year), and a random intercept effect (elevation-specific population nested within species nested within genus) to account for variations in size and size change both within and between species and along elevation. The body sizes of individual species decreased by 1.3% on average (mean \pm SE shrinkage = 0.25 ± 0.04 mm; $t = -6.37$, $P < 0.00$). (now line 139). Note that the overall slope estimates were not significantly different between the two approaches.

Following these revisions, we have revised the original figure and used a histogram to demonstrate the number of species that show different forewing length changes. We have moved this figure to supplementary material (now Figure A1.1), allowing space in the main text to include the conceptual scheme (now Figure 1) and data table (now Table 1), as suggested by reviewer #2.

We also add a new paragraph to discuss the complexity of size change, whilst recognizing we find a significant reduction. We discuss phenotypic plasticity, evolutionary adaptation, the direct effect of temperature and indirect effect of host plant quality, and possible interplay between original size and number of generations since the re-survey (now line 159-169).

We also compare the size changes with temperature that we report with other studies, by transforming wing length to dry weight, following Garcia-Barros (2015). This produces a decline of 2.8% on average or $4\%^{\circ}\text{C}^{-1}$, based on 0.7°C warming. This magnitude of reduction is higher than $-1.43\%^{\circ}\text{C}^{-1}$ for terrestrial ectotherms (Forster et al. 2012) or $-0.35\%^{\circ}\text{C}^{-1}$ estimated by laboratory warming (Horne et al. 2015). Please see line 169-177 for full discussion.

4. Figure 3 was a little unconvincing. It would be useful to see the range dynamics laid out according to body size class. That would provide a much stronger line of support for the size related range boundary change contention than the current figure. Again, it is not possible to see how much non-independence is introduced by species that occur over multiple sites. Indeed, the concluding sentences about the significance of richness should be considered in light of the role of species that are widespread across the elevational extent compared with those that are not. Without dissecting out this pattern, it is difficult to consider the outcomes reliable.

Our response:

Figure 2c shows that colonizing species are disproportionately small-bodied species – at the four highest elevation sites. In contrast, our new analysis finds no correlation between initial body sizes and range shifts (Figure A2.2 – there is simply a larger pool of smaller bodied species at lower elevations available to colonise upwards). A related issue is that there could be a trade-off (i.e. non-independence) between range shift and size shrinkage, as reviewer #3 pointed out; but there are no correlations between the magnitude of size changes and range shifts (Figure A2.3).

The effects of widespread species on assemblage size changes have been discussed in point # 1 and the discussion has been revised accordingly line 208-218.

The complex interplay of biotic and abiotic factors along the mountain gradient may explain the idiosyncratic responses of size and range shifts among species. We add context in the introduction (please see line 44-51) and relevant discussion on the size dependent range shift in lines 188-198.

5. The authors would be well served by considering, again, the established literature on size patterns as opposed to the mechanisms explaining them. Bergmann's rule, for example, has several mechanistic explanations, and for ectotherms these take a

variety of forms. Understanding what the case is within the more widely distributed relative to the narrowly distributed species might help to disentangle these effects. The intraspecific analysis could be usefully adapted to do so. That would provide perhaps more insight than the loosely associated discussion of Bergmann's rule, the temperature-size effect and related concepts. The discussion does not really provide clear evidence of the mechanisms the authors have in mind beyond some general notion that temperature affects ectotherm size. Although the material in Table A2.1 and Figure A2.1 (but also remembering that the altitudinal variation figures is an unspecified mix of intraspecific and interspecific information) go some way towards doing so the general discussion really does not provide the kinds of insights the data would allow. Moreover, these data also suggest that partially out the impacts of widespread versus localized species might be useful.

Our response:

Thank you – we have revised and improved the manuscript. We have increased the number of citations, added relevant hypotheses and included some further discussion. We add new analysis (above) and explanations to reflect the complexity and diversity of altitudinal size clines, confounding factors affecting size changes in mountain assemblages, and the non-independence of size patterns due to variation in species' range size.

Reviewer #2 (Remarks to the Author):

Wu and co-authors use wing-length data for over four thousand moths collected across an elevational gradient on Mt. Kinabalu, Borneo, in 1965 and 2007 to provide what they claim is the first empirical evidence of within-species reductions in body sizes for tropical insects. The unique dataset allows the authors to separate direct effects of reduced body size from indirect effects on assemblage size structure caused by geographic range shifts. Overall I found the paper to be well written, and mostly clear in its description of the problem and the author's strategy to address the problem. This really does appear to be quite novel use of the data, and I enjoyed reading the manuscript. Below I have listed some general comments I have about the paper, followed by more specific comments that I think could be more easily addressed.

General comments:

My main criticism of the paper is its lack of attention to other possible causes of the

observed patterns. I get that the temperature increased on average 0.7C over the past 42 years, but how do you know the changes in moth size were driven by this change? The only time I can see reference to other possible factors is on lines 139-140, where you say there was 'limited habitat change', but if you want to lay out these results as a direct consequence of climate warming then these issues need to be address.

Our response:

The referee is correct that we should add further explanation. We now add the following description from line 84. "Chen et al. (2009)⁹ resurveyed the moth transect on Mt. Kinabalu in 2007, 42 years after the original study in 1965⁴², using the same field protocols and visiting the same sites along the elevation gradient. Since Mt. Kinabalu was established as a Malaysian national park in 1964, the habitat remains largely undisturbed. We were thus able to compare the range shifts^{9,43} and body size changes (this study) under climate warming with limited confounding factors. Over the 42 years' study interval, moth species moved uphill by 67m on average, in response to 0.7°C warming, reshuffling the species composition of assemblages along the transect."

In a similar vein, what is the possibility that your observed results are highly dependent on the specific time interval you analyzed, and if you had analyzed assemblage size structure over a different time interval then could you have observed an opposite effect? Trait distributions in natural communities can respond very quickly to environmental change, and so moth wing lengths may shift on a much smaller time frame than the 42 years you actually observed. The work by Rosemary and Peter Grant on Darwin's finches is an excellent example of how quickly trait distributions in local communities can fluctuate over quite short time frames.

Our response:

Thank you for raising the issues of environmental stochasticity and rapid species responses. Please see our reply to reviewer # 1, point 1. We have now discussed this in lines 202-206.

I couldn't find details about how you obtained your climate data – was it predicted from climate models, or based on direct measurement of temperature (i.e., weather station)? Do you have climate data for each site? If so, can you explore temperature changes at each site with their relative change in moth size, to get a more nuanced picture of how temperature relates to changes in size structure?

Our response:

The climate data were derived from the Global Historical Climatology Network (Chen et al. 2009). Unfortunately, long term site-specific climate data do not exist for Mt. Kinabalu.

I think the Abstract needs work. The description of the problem is not as well laid out as in the first paragraph of your Introduction. I think restructure the your Abstract to be more in line with this text.

Our response:

We have reworked the abstract, in line with the introduction.

Have you thought about moving Figure A1.1 to the main text? I think it would clarify quite a few non-intuitive points for the reader (especially the boundary/non-boundary metrics you explore). If you have space in the main text, then you could either move do it in your results section as is (~Line 100), or perhaps better would be to modify the figure to make it a more general introduction to the effects of species redistribution on size structure even earlier when you first talk about these issues (~Line 52-53).

Our response:

We have moved Fig. A1.1 to the main text (now as Figure 1) and modified it to better demonstrate the conceptual framework of our study.

Lastly, perhaps I missed it, but I didn't find anything about how the moths were sampled? I think this is important, just to confirm that sampling was comparable across the two samples.

Our response: We have added more details as described above, lines 84-90.

Specific comments

Line 20: Be specific with what you mean by 'reduce the body sizes of animal species' – do you mean individuals, populations, cohorts, mean community size etc? or all of the above? This becomes clear after reading the paper, but it should be clear from the outset.

Our response:

We now use 'intraspecific size change' throughout the manuscript when describing changes to the average body sizes of individuals within a given species.

Line 25: How many species/taxa?

Our response:

We were able to collect forewing length data from 5536 specimens of 277 species in the historical survey, and 3053 individual specimens from 219 species in the later survey, representing 74.4% to 99.1% of individuals at a site (lines 92-94, Table 1).

Line 30: Over what time period was this 4.9%?

Our response:

There were 42 years between the two surveys. Information added (line 29).

Line 48: I'd change 'are likely' to 'appear to be', if that's what previous research shows.

Our response:

Done.

Line 52-53: The second sentence of this paragraph needs to be more clearly linked to the first. Do you mean in general, or for terrestrial systems only?

Our response:

The main text has been restructured to better reflect the knowledge gaps and hypotheses we test (Lines 44-54).

Line 61: I think you need a dedicated sentence describing how climate of the region has changed between surveys, before you say how much moth assemblages were altered.

Our response:

Done. We have revised the main text (Lines 84-90) to better reflect the study background.

Line 84-85: You keep saying the site names, but they aren't shown in your Fig 1 so it seems a little redundant. If they aren't necessary, could just call them 1 through 7, or just by their elevations.

Our response:

We have added elevations to the site names, and in Table 1, which summarizes resurvey information so that it will be easier for readers to follow.

Line 88-91: This is for the entire dataset, but did you find similar results within sites, or you just don't have the power to explore within-site patterns for specific species?

Our response:

To obtain a sufficient sample size for meaningful comparisons for each species, we have to analyze species at all sites together. Thus, within-site patterns for each species are included in the linear mixed model as a random effect but are not reported separately.

Line 89-90: Its very difficult to see this 1/3% change from the current figure format – is there not a better way to present these data to better highlight the pattern of interest?

Our response:

The 1.3% reduction comes from the Linear mixed model. We agree that the original Figure 2 was not that informative and a new histogram of frequency of changes in forewing length is added to Figure A1.1.

Line 114: Climatically warmer?

Our response:

The main text has been revised.

Line 128-130: So what are the consequences of this for global responses to climate change – e.g., are colder and less diverse high-latitude habitats more likely to have their size structure altered?

Our response:

Yes, we believe so from our findings, and point this out in line 221-225: “Low

species diversity at higher elevation sites rendered them particularly susceptible to the impacts of species- and size-restructuring under climate change, given that the immigration of a relatively small number of small-bodied species could alter the overall assemblage profile (Fig. 2c). ”.

Line 136-142: Some of this text seems to be irrelevant to what I was expecting in here – a description of where you got the moth data, and how that data was collected in the field.

Our response:

We have deleted this text and explain the moth transect resurvey elsewhere in the main text (Lines 84-90).

Line 179-180: I think mention this in the main text, it’s a pretty important caveat to highlight for the reader. I might also add text from Lines 182-184, stating that using pres-abs or abund data didn’t change results qualitatively.

Our response:

Many thanks for pointing this out. We have moved these sentences to the main text (Lines 111-115).

Line 212-221: Again, I think some of this could be moved to the main text, perhaps even the figure.

Our response:

We now consider this in the main text. We have moved (and improved) Figure 1 to main text, as suggested. We also explain more fully how we disentangle community composition effects associated with range shifts from intraspecific body size changes in the main text. Please see line 97-107.

Figure 2: I’m not sure this is the best format to show whether on average species sizes get bigger or smaller – its tough to see whether more points lie above or below the red line.

Our response:

The new Figure A1.1 is easier to interpret.

Figure 3: In your caption I think your reference to Table A1.3 for panel a) should be

Table A1.2, and panel b) should be Table A1.3?

Our response:

Many thanks for pointing this out. Revised accordingly.

Table A1.1 – If you had space, I'd think of moving this table to the main text – its pretty important to the story. Can you also add what the temp changes were at each site (or you only have entire region?).

Our response:

Changed accordingly. It is now Table 1 in the main text. We only have estimated warming for the entire region.

Figure A1.1 – This figure is quite relevant – do you have space in the main text for it?

Our response:

Changed accordingly. It is now Figure 1.

Reviewer #3 (Remarks to the Author):

This study appears to build on an earlier publication by the same research group (Chen et al. 2009. PNAS 106: 1479-1483), which reported long-term (1965-2007) elevation increases in moth assemblages along an altitudinal gradient in Borneo. Here the authors extend their previous work, examining how the mean body size of moth assemblages has changed over time. Importantly, they investigate the relative contribution of range shifts and intra-specific body size reductions to the observed patterns.

At the assemblage level, the authors observe significant reductions in the mean size of moths. These reductions appear to be in response to climate warming (+0.7°C over a 42 year period), primarily driven by range boundary shifts (smaller species moving towards higher colder altitudes), and to a lesser extent body size reductions within species.

I agree that integrating data on species redistribution with physiological responses to temperature (i.e. intra-specific body size reductions) is important for understanding how organisms will respond to climate warming (not forgetting shifts in phenology too). These types of studies are underrepresented in the literature, probably because

we lack the necessary long-term (decadal-scale) data to examine such patterns. Consequently, I think this is a valuable study.

That being said, there are several areas in which the manuscript could be strengthened, as well as some caveats that should be better acknowledged. I outline these below.

- The paper would benefit from restructuring. In particular, I feel it is lacking an explicit set of testable hypotheses. Only after their results (line 111) do the authors predict how the direction of body size clines along altitudinal gradients (e.g. Bergmann's rule) will influence shifts in assemblage size structure with warming. Given the predominantly positive altitudinal-size clines on Mt. Kinabalu (i.e. smaller species and/or individuals generally found at warmer, lower elevations), the authors can make some reasonable predictions about how range boundary shifts are likely to contribute to changes in mean assemblage body size. In contrast, where altitudinal-size clines are largely absent, one might predict that range shifts may be less influential in driving changes in assemblage size structure. The authors could do a much better job of setting up these broader hypotheses, before describing how they use moths on Mt. Kinabalu as a case study to test them.

Our response:

Thank you for the constructive suggestions. We have strengthened the introduction to include this information, and outline the hypothesis and predictions that we test (Lines 44-51).

We have also restructured Appendix 3 in order to show the basis for our predictions and general hypothesis more clearly through analyses at both inter- and intra-specific level. This was stimulated by historical observations that many species-groups and even genera appear to show a Bergmann cline, particularly within the subfamily Larentiinae that is predominant in the four uppermost sites.

- Long-term patterns in body size can be strongly confounded by variation in life stage and age structure between years. For this reason, it is important to measure the same size-at-stage across years. The life cycle of Lepidoptera make them particularly suited for avoiding these issues, especially when sampling the winged adult form. Nevertheless, this is a very important methodological consideration that should be highlighted to the reader - especially if others intend to investigate these issues in other species using preserved collections.

Our response:

We highlight this consideration in the first paragraph of methods: " For pinned Lepidoptera specimens, forewing length is the best preserved trait that is strongly correlated with overall body size 49 and enables same-stage (i.e. adult) comparisons to be made across years" Please see lines 241-243.

- It would be useful to comment on the average temperature gradient along the elevation transect. Whilst moth species moved uphill by 67m on average, how does this compare to the gradient in temperature? To what extent do these range shifts compensate for the 0.7C of warming? I see some of this was briefly discussed towards the end of their earlier publication (Chen et al. 2009. PNAS 106: 1479-1483), but it is particularly important here. Though not the biggest contributor, intra-specific reductions in size were still observed – the persistence of this physiological response would suggest that range shifts do not fully compensate for the increase in temperature. It's interesting that species aren't shifting their distribution to completely offset the physiological effects of warming. Why might this be the case? This should form a more important part of the Discussion.

Our response:

The magnitude of range shifts of moths on Mt Kinabalu do not appear to fully compensate for the degree of warming. We add this information (Lines 188-198) as context for discussing the interaction between two biological responses of range shifting and intra-specific size changes. Please see next paragraph.

- Following from the above, did species that exhibited smaller range shifts exhibit stronger intra-specific reductions in body size? Is there a trade-off between these two responses; are they negatively correlated? This type of analysis would be really valuable and particularly novel.

Our response:

Good question. Our analyses do not find a significant correlation between range shifts and size change (Fig. A2.3). Nor did we find any evidence for size-dependent range shifts (Fig. A2.2).

We have included further consideration of possible trade-offs, altitudinal clines, and unresolved issues at: lines 188-198, 230-233.

- Figure 2: The authors provide a 1:1 reference line, but I'd also like to see a

reduced major axis (RMA) regression through the data (with information about the intercept and slope). This would be particularly interesting given that intra-specific size changes have been shown to vary systematically between species based on key life history traits, including body size. Eyeballing the data, it looks like smaller species might show stronger reductions in size – was this tested? Similarly, do the species in the data set vary in generation time? Previous work on arthropods has shown that smaller multivoltine terrestrial species often reduce their size with warming, whilst larger univoltine terrestrial species generally increase in size (likely driven by a concurrent increase in season length).

Our response:

In order to address these concerns from all three reviewers, we have replaced the original Figure 2 by a histogram, and moved the original figure to the appendix (Figure A1.1). We found no relationship between species' initial sizes and size change (Figure A2.1). We do not have generation time data for the geometrids, but we have now included some discussion of the importance of generation times on our findings (Lines 164-168).

- Line 195: Here the random effect is on the intercept only, but not the slope. As I understand it, this accounts for variation in the mean size of species, but assumes that the rate of change (i.e. the slope) between years does not vary between species. Relating to my comment above, did the authors consider investigating how these slopes varied between species? Are there any systematic differences in the magnitude of range shifts and intra-specific size reductions between species? Incorporating these kinds of questions/analyses could really enhance the work.

Our response:

Please see our reply to Reviewer 1, point 3. We conducted additional analysis to include both intercept and slope as random factors, but the estimated random slope variation ($= 0.043 \pm 0.21$) was not significantly different from zero among species.

In the main text, we report the model outputs, including a fixed effect (year), and a random intercept effect (elevation-specific population nested within species nested within genus) to account for variations in size and size change both within and between species, and with elevation.

- It is nice to see that the authors acknowledge and test the robustness of their

methodology, in particular with regards to the estimation of missing data (Appendix 3). However, the authors could do much more to acknowledge the limitations of their study in the main text. Whilst the size of the data set is considerable, ultimately the conclusions are derived from 2 points in time (1965 and 2007). This makes it difficult to tease apart the potential drivers, as we cannot tell how closely assemblage size structure has tracked changes in environmental variables over time (namely temperature but also other variables). I appreciate this is the nature of the available data, but this needs to be better acknowledged.

Our response:

We now acknowledge the limitations and rationale of our study more clearly (line 202-206): "A possible concern in this study is that we compare two time snapshots, and it would be desirable to have multi-year data in future studies. This issue is partly mitigated by the long time period between the two surveys 9,56 but, nevertheless, data for additional years and other research systems are needed to determine the generality of our findings."

- Be careful when emphasizing novelty (e.g. line 26). Consider rephrasing, as intra-specific body size reductions in tropical insects have been investigated in the lab, across latitude and intra-annually across seasons. Here the novelty seems to come from the investigation of long-term (decadal) changes in size. Be more explicit.

Our response:

We have revised the entire abstract and this section now reads: " We found significant size reductions for tropical insects (a mean shrinkage of 1.3% per species over 42 years)."

Overall this is an interesting study. I hope my comments are constructive and help to improve the work.

Our response:

We greatly appreciate the comments, which have helped us to improve our manuscript.

REVIEWERS' COMMENTS:

Reviewer #1 (Remarks to the Author):

Thanks for such clear, critical and rigorous responses to the concerns raised, including new and convincing analyses. The work done has benefitted the ms considerably. The ms makes an important, novel and engaging read. The concluding sentence is especially important!

I have only two remarks which the authors might wish to consider.

1. On line 58 the statement that reductions in size lead to an increase in metabolic rates can be interpreted in two ways, and one of them is wrong. Indeed, as written, I would say the wrong interpretation is likely. On either an individual or a species basis, body size reduction leads to lower metabolic rates. That's actually what all empirical scaling data show, and the reason is obvious – there's less tissue that is converting resources to ATP to do work. Reductions in body size do lead to increases in mass-specific metabolic rate. But that's actually something of a different matter. I would strongly suggest that the authors point out that reductions in size lead to reductions in rate. And that's what equations 1 and 3 of Ref 27 (Gillooly et al. 2001) show. They might prefer to say reductions in size lead to increases in mass-specific metabolic rate, but then ref 27 is not entirely appropriate.

2. The key outcome line of the introductory paragraph reads thus: 'We found significant size reductions in tropical insects (a mean shrinkage of 1.3% per species over 42 years), but range shifts caused most size re-structuring of assemblages, due to uphill shifts of relatively small species, especially at high elevations.'

When one examines Figure 1 though, whether or not the authors intended this, one immediately comes away with the impression that bigger moths have moved into higher elevations in 2007. I have stared at this figure for a fair while and could not get my mind to change.

Thus, the authors might want to rethink this figure.

Reviewer #2 (Remarks to the Author):

The authors have largely addressed the previous concerns I had with their paper, to where the manuscript is now clear in their presentation of an interesting and important set of ideas and data that will be of broad interest. Below are several mostly minor suggestions that I think might improve the manuscript.

Line 28: Having the word 'tropical' is awkward here - makes me start to wonder about latitudinal effects (which you don't explore at all). I'd remove, or move to another sentence.

Line 34: Are you referring to the 0.5% and 0.6% effects? It's not currently clear to me.

Line 39: I don't like the word 'individualistic' here, as certainly there are some general rules emerging (as your current paper shows). You could just replace it with 'variable'

Line 44: Are you not also talking about changes to the size of species as well?

Line 89-90: Is there a reference for this?

Line 98: In the caption for Fig. 1 you refer to three main processes - but you just list two here. It's confusing, so try to be consistent where possible.

Line 98-102: What the four categories of species composition changes are is unclear from the text to me - sentence needs reworking to make clear.

Line 104-105: Moving this to earlier in the paragraph - where you can start by stating that there are a total of 6 processes - might help clarify the issues I identify in the above couple of comments.

Line 109-111: Ditto to my previous comments. This just adds to the confusion. I might integrate above, or at minimum rework all this text to be clearer about how you are defining/categorizing processes (and how it links to Fig. 1)

Line 118: Can you point to some Figs/Tables in the supp info that support this statement?

Line 124: I think refer to Fig. 2a at the end of this first sentence, and maybe remove it from the next sentence. I think do this because it is difficult to estimate the changes at each elevation from the data as presented in Fig. 2a.

Line 128-129: I think delete 'where the average body sizes of assemblages were reduced by 0.79 mm to 2.13 mm 128 (-4.6% to -11.7%)' as its mostly repetitive from the previous sentence.

Line 133-136: While I like the idea of Fig. 2c, I'm not sure it's the best way to highlight the trend you describe here - it's very tough to tell this trend from the way the data is presented.

Line 140: Personally, I'd like to see Fig. A1.1 in the main text

Line 142: I'm confused, shouldn't this 4/9% be 0.6% (which i got from your abstract). If I am confused, then it's probably because of your inconsistency is how you refer to different processes/sets of processes through the manuscript.

Line 156-157: Delete "over the 42-year period along the 156 elevation transect."

Line 164: Maybe also or instead say 'ecological interactions' (just to be broader than plant quality, as it could also be predators, or parasites, or competition, etc.)

Figure 1: This figure is a good addition, but I think it could be made clearer and more complete. What about a third column on the right that explains the effect (at each elevational band) of the process on assemblage size structure? It would also be great if you could add the three processes you mention in the caption to the figure (i.e. intraspecific size change, boundary shifts, and within range sizes).

Reviewer #3 (Remarks to the Author):

The authors have done a reasonable job of addressing the issues raised. I have no further major comments.

EDITORIAL REQUESTS:

[1] Further to the referees' comments, I would like you to check whether your usage of the Bergmann's rule terminology in the text is consistent with current practice. I am aware that the interpretation of the original rule has changed through time, but to my understanding most ecologists nowadays interpret it as: colder conditions beget larger-bodied individuals within a species, not (necessarily) larger-bodied species. If this is correct, it appears in contrast to what you state (L46-47, L613). Could you have a second look at those statements and amend or contextualise as necessary? Consider for instance <https://onlinelibrary.wiley.com/doi/abs/10.1111/j.1469-185X.2009.00097.x>, and <https://onlinelibrary.wiley.com/doi/full/10.1046/j.1472-4642.1999.00046.x>

Author response [1]: Many thanks for this comment. We are aware of the wide application of Bergmann's rule and the lack of consensus on which taxonomic level, e.g. intra- or inter-specific, relationships with Bergmann's rule are examined. It is possible that intra- and/or inter-specific relationships with Bergmann's rule may contribute to the assemblage size-structuring we observe, and so we investigated patterns at both intra- and inter-specific levels (Supplementary Table 3 and Figure 7). We now cite Blackburn et al. (1999) and have revised the sentence to make this clear (L42-46):

"If Bergmann's rule is operating at interspecific and/or intraspecific levels, then larger species and/or individuals are found towards higher latitudes and/or altitudes¹⁰⁻¹⁵, such that range shifts associated with climatic warming may generally reduce the average size of new communities, as small species/individuals colonize and larger species/individuals disappear."

[2] TITLE PAGE

*** I slightly reworded the title to avoid excessively strong causal language, please check whether you agree with the text below. If you would like to suggest an alternative, please ensure that it does not exceed 15 words and does not contain punctuation.**

« Reduced body sizes in climate-impacted tropical insect assemblages are primarily explained by range shifts »

Author response [2]: Revised as suggested.

[3] * Please label the “Introductory paragraph” as Abstract.

Author response [3]: Revised accordingly.

[4] *The current abstract is significantly longer than the 150-word limit, so I drafted a shortened version, please let me know whether you approve it. You are welcome to edit it; if so, please bear in mind the word limit, discuss the current work using the present tense, and do not include references.

« Both community composition changes due to species redistribution and within-species size shifts may alter body size structures under climate warming. Here we assess the relative contribution of these processes in community-level body size changes in tropical moth assemblages that moved uphill during a period of warming. Based on resurvey data for seven assemblages (>8000 individuals) on Mt. Kinabalu, Borneo in 1965 and 2007, we show significant wing-length reduction (mean shrinkage of 1.3% per species). Range shifts explain most size re-structuring, due to uphill shifts of relatively small species, especially at high elevations. Overall, mean forewing length shrank by ca. 5%, much of which accounted for by species range boundary shifts (3.9%), followed by within-boundary distribution changes (0.5%), and within-species size shrinkage (0.6%). We conclude that the effects of range shifting predominate, but considering species physiological responses is also important for understanding community size reorganization under climate warming. »

Author response [4]: Thank you very much for the concise re-writing. We revised accordingly.

[5] MAIN TEXT

*** Please divide the main text into the following sections: Introduction, Results, Discussion, and Methods, each of which must begin with a heading. Results and Discussions can also be combined. If the Discussion is a separate section, please do not divide it into subsections.**

Author response [5]: We have re-organized the main text as requested. For clarity, we divide the Results into two subsections and have added appropriate subtitles, but can delete these if preferred. We also revise and move L96-107 to L123-134 and revise and move L109-118 to L153-167 to explain the results more clearly.

[6] * The final paragraph of the Introduction should summarise the major results and conclusions of this manuscript, in the present tense. I suggest that such paragraph can be made using the text from L72 to L97, plus a couple of sentences to give more information on the findings and conclusions.

Author response [6]: We revise the end of introduction by using text from L72 to L97 as suggested, and added more sentences as follows (L79-86):

“We find that over 42 years, moth body size, in terms of fore-wing length, reduces 1.3% on average for species and ca. 5% for assemblages. Positive body size clines along the elevational transect are observed at subfamily levels and thus species range shifts contribute to the reduction in assemblage size at higher sites. Overall, the assemblage size shrinkage is driven mainly by species range shifts (3.9%) and to a much lesser degree by within-species size changes (0.6%). Range-shift induced species reshuffling brings substantial size re-structuring, and assemblages of low biodiversity are particularly susceptible to impacts of range shifting under climate change.”

[7] LANGUAGE AND STYLE

*** Please remove “the first” in L159. We do not allow language such as "new", "novel", "for the first time", "unprecedented", etc., novelty should be clear from the context.**

Author response [7]: Done, and we have checked that we do not use these terms anywhere else in the manuscript.

[8] * We do not allow inferences based on data that is not present in the manuscript or not published. Please back up statements in L118 and L277 with supplementary figures and data.

Author response [8]: We (i) repeated the analyses by weighting species according to their relative abundances, (ii) added Supplementary Figure 3 and 4 (see below) and (iii) added text to explain the results as follows (L153-167):

“We repeated these presence/absence analyses by weighting species according to their relative abundances at each site in each year. The results were comparable to those obtained using presence/absence data, with average sizes of moths in the

assemblages shrinking at five out of seven sites (Supplementary Figure 3a) , and coefficient of variation results similar except at the two highest sites (Supplementary Figure 3b). Dividing the effects into different processes, distribution margin changes still reduced overall assemblage body sizes (a robust result), but there was also a major contribution of non-boundary dynamics (primarily via abundance changes of small versus large-sized species) in reducing assemblage size profiles (Supplementary Figure 4a). These tended to increase the within-community coefficient in variation (except for the highest sites; Supplementary Figure 4b).”

Supplementary Figure 3 Moth assemblage size structure in 1965 (black) and 2007 (red) based on weighting species by their abundances at each site. (a) Average forewing length (mm). (b) Coefficient of variation of species forewing length. (c) Frequency distribution of species forewing length. In (a) and (b), mean and 95% confidence intervals at each site are shown, based on 500 resamples. Data points are overlaid. Asterisks indicate significant ($p < 0.05$) differences between 1965 and 2007. In (c), number of species are on log10 scale and overlaps between the two years are illustrated in grey.

Supplementary Figure 4. Contribution of range boundary shifts (black), intraspecific size change (white) and non-boundary dynamics (grey) to changes in moth assemblage size structure from 1965 to 2007, based on weighting species by their abundances at each site. (a) Change in average species forewing length. (b) Change in coefficient of variation of species forewing length. Asterisks indicate components with effect sizes significantly different from zero at the 95% confidence level, based on 500 resamples.

We excluded the sentence in L227: “High species diversity may thus buffer the impacts of range shifting on assemblage functional traits” as it is very broad and may lead to confusion.

[9] * Please do not use italics or bold font to convey emphasis (in both the main text and the display items).

*** Please avoid using speech marks around words or phrases. In most cases they are unnecessary.**

Author response [9]: We have excluded italics, bold font and speech marks in the main text and tables. We have retained bold font in Supplementary Tables 1 & 2 to show significant values.

[10] METHODS AND DATA

*** The Methods section subheadings should each be fewer than 60 characters in length, spaces included.**

Author response [10]: Confirmed.

[11]* Please include in the Methods section a statement affirming that you have complied with all relevant ethical regulations for animal testing and research. Please also include a statement explicitly confirming if the study received ethical approval, including the name of the board and institution that approved the study protocol (or a statement that the study was exempt from requiring ethical approval).

Author response [11]: We have added this sentence(L270-271):

“Note that the study was exempt from requiring ethical approval for animal testing and research as it works on preserved specimens of insects.”

[12] * All *Nature Communications* manuscripts must include a section titled "Data Availability" as a separate section after the Methods section and before the References. For more information on this policy, and a list of examples, please see <http://www.nature.com/authors/policies/data/data-availability-statements-data-citations.pdf>

*** DATA SOURCES: We strongly encourage authors to deposit all new data associated with the paper in a persistent repository where they can be freely and enduringly accessed. We recommend submitting the data to discipline-specific, community-recognized repositories, where possible and a list of recommended repositories is provided here: <http://www.nature.com/sdata/policies/repositories>**

*** If a community resource is unavailable, data can be submitted to generalist repositories such as figshare (<https://figshare.com/>) or Dryad Digital Repository (<http://datadryad.org/>). Please provide a unique identifier for the data (for example a DOI or a permanent URL) in the "Data Availability" section, if possible. If the repository does not provide identifiers, we encourage authors to supply the search terms that will return the data. For data that have been obtained from publically available sources, please provide a URL and the specific data product name in the "Data Availability" section. Data with a DOI should be included in the reference list and cited where relevant.**

Please refer to our data policies here:

<http://www.nature.com/authors/policies/availability.html>

*** In an effort to ensure reproducibility of research data, we now also require that you provide a separate source data file. The source data file should, as a minimum, contain the raw data underlying all reported averages in graphs and charts, and uncropped versions of any gels or blots presented in the figures. To learn more about our motivation behind this policy, please see <https://www.nature.com/articles/s41467-018-06012-8>.**

(see more information about the Source Data file below under the heading SUPPLEMENTARY INFORMATION)

Please add a reference to the source data file in the "Data Availability" section. For example:

"The source data underlying Figs 1a, 2a–d, 6d, h and 7c and Supplementary Figs 1a and 5d are provided as a Source Data file."

Author response [12]: As the results are generally based on re-sampling the raw data, the source data for figures and tables are the original raw data. Our Data Availability statement is:

"The source data to generate all figures (except conceptual Figure 2) and tables in the main text and the supplementary information are available in Figshare with data DOI: 10.6084/m9.figshare.9728411."

[13] END NOTES

*** Please supply an "Author Contributions" section after the Acknowledgement section that refers to all authors.**

Author response [13]: We add Author Contributions section as follows:

“C.-H. Wu, I.-C. Chen, and C.-K. Ho designed the study. C.-H. Wu measured the samples and analyzed the data. J. D. Holloway provided taxonomic guidance and input to Supplementary Table3 and Note 2. C.-H. Wu, I.-C. Chen, and C.-K. Ho prepared the manuscript with editing, comments and interpretation from J. D. Holloway, J. K. Hill, and C. D. Thomas.
”

[14] * Please provide a "Competing Interests" section after the "Author Contributions" section that refers to all authors. If there are no competing interests, please add the statement "The authors declare no competing interests."

Author response [14]: We declare no competing Interests.

[15] * Please ensure the references are numbered in the order they appear in the text, followed by the tables and figures.

Author response [15]: Confirmed.

[16] DISPLAY ITEMS

*** Please check whether your manuscript or Supplementary Information contain third-party images, such as figures from the literature, stock photos, clip art or commercial satellite and map data. In particular, please indicate whether you have permission to use the photograph and moth shapes in Figure 1. We strongly discourage the use or adaptation of previously published images, but if this is unavoidable, please request the necessary rights documentation to re-use such material from the relevant copyright holders and return this to us when you submit your revised manuscript.**

Author response [16]: We have added text to the Acknowledgements to credit photographs (L366-368) and the permission for the modification of moth photo in Nature Communications has been uploaded.

“The photo of Mt. Kinabalu in Fig. 2 is taken by I.-C. Chen; the moth silhouette is modified from a photo taken by S. Wu.”

[17] * Where p-values are presented as symbols/letters, please ensure that these are defined in the relevant figure legend, and the statistical test used to generate

them is stated.

Author response [17]: We checked the manuscript and confirm that p-values and their associated statistical tests are explained in the text.

[18] * Please overlay the corresponding data points (as dot plots) in panels a and b of Fig. 2, and in supplementary figures A4.1, A4.2, A5.1 (panels a and b), and A5.3 (panels a and b).

Author response [18]: We have overlaid the data points in these figures, now numbered Fig. 1a, 1b., Supplementary figure 10, 13, 14 and 16.

[19] * Data in tables must be free from bold/italic formatting unless this has been clearly defined in the footnote. Tables need to be black and white, fit onto a single A4 portrait page and can contain only one row of column titles. Finally, we are unable to merge cells or include vertical dividing lines or diagonal lines. Please format your tables accordingly.

Author response [19]: Revised accordingly.

[20] * Please define any new abbreviations, symbols or colours present in your figures in the associated legends. Please do not use symbols in your legend, instead please write out the symbols in words (blue circles, red dashed line, etc.).

*** In each figure and supplementary figure where error bars are used, they must be defined. One statement at the end of each figure is sufficient if the error bars are equivalent throughout the figure.**

Author response [20]: Revised accordingly.

[21] SUPPLEMENTARY INFORMATION

*** We do not edit Supplementary Information files; they will be uploaded with the published article as they are submitted with the final version of your manuscript. Any tracked changes should be removed from the file and the file should be provided as a PDF file. Supplementary Figures do not need to be provided separately.**

*** Please supply your Supplementary Information as a separate PDF file, not within**

the manuscript file.

*** Please replace general citations to the Supplementary Information (e.g. "see Supplementary Information") with specific citations (e.g. "See Supplementary Figure 1", etc.).**

*** Please rename the Appendix as Supplementary Information. The only section headings permitted here are Supplementary Figures, Supplementary Tables, Supplementary Methods, Supplementary Notes (must be numbered), Supplementary Discussion, Supplementary References. All other section headings and numbering should be removed or relabelled.**

*** In the Supplementary Information file and the main manuscript text, supplementary items must be labelled and cited using only the following formats: Supplementary Figure 1, Supplementary Table 1, Supplementary Methods, Supplementary Note 1, Supplementary Discussion, and Supplementary References. Please note the use of "Supplementary" and that we do not use the "S" prefix.**

*** Please note that we do not allow panels in Tables or Supplementary Tables – please relabel/renumber accordingly.**

*** Supplementary References should appear at the end of the Supplementary Information file, and should be self-contained and numbered from 1. References mentioned in both the main text and the Supplementary Information should be part of both reference lists so that the Supplementary Information does not refer to the reference list in the main paper and vice versa.**

Author response [21]: We have re-organized all the supplementary information according to these instructions. Supplementary information is now cited in the correct format in the main text. Panels have been removed and references are shown at the end of the file. We have submitted the supplementary information as a separate PDF file

[22]* Please provide the Source Data file(s).

*** Within the Source Data file, each figure or table (in the main manuscript and in the Supplementary Information) containing relevant data should be represented by a single sheet in an Excel document, or a single .txt file or other file type in a zipped**

folder. Blot and gel images should be pasted in and labelled with the relevant panel and identifying information such as the antibody used. We also encourage you to include any other types of raw data that may be appropriate. An example Source Data file is available demonstrating the correct format:

<https://www.nature.com/documents/ncomms-example-source-data.xlsx>

The file should be labelled "Source Data", with the title and a brief description included in your cover letter, and should be mentioned in all relevant figure legends using the template text below:

"Source data are provided as a Source Data file."

Author response [22]: We have prepared the source data accordingly and uploaded the raw data to Figshare with data DOI: 10.6084/m9.figshare.9728411.

[23] REVIEWERS' COMMENTS:

Reviewer #1 (Remarks to the Author):

Thanks for such clear, critical and rigorous responses to the concerns raised, including new and convincing analyses. The work done has benefitted the ms considerably. The ms makes an important, novel and engaging read. The concluding sentence is especially important!

I have only two remarks which the authors might wish to consider.

1. On line 58 the statement that reductions in size lead to an increase in metabolic rates can be interpreted in two ways, and one of them is wrong. Indeed, as written, I would say the wrong interpretation is likely. On either an individual or a species basis, body size reduction leads to lower metabolic rates. That's actually what all empirical scaling data show, and the reason is obvious – there's less tissue that is converting resources to ATP to do work. Reductions in body size do lead to increases in mass-specific metabolic rate. But that's actually something of a different matter. I would strongly suggest that the authors point out that reductions in size lead to reductions in rate. And that's what equations 1 and 3 of Ref 27 (Gillooly et al. 2001) show. They might prefer to say reductions in size lead to increases in mass-specific metabolic rate, but then ref 27 is not entirely appropriate.

Author response [23]: Many thanks for pointing out this. We now realise that the sentences are misleading as written. We have revised the text as follows (L54-58):

“For ectothermic organisms in particular, body size reduction is often associated with faster developmental rates at higher temperatures, in accordance with temperature–size rules^{26,27}. Body size reduction may also be related to higher metabolic rates under warming, if the increasing metabolic cost is not compensated by higher food intake²⁵”

[24] 2. The key outcome line of the introductory paragraph reads thus: ‘We found significant size reductions in tropical insects (a mean shrinkage of 1.3% per species over 42 years), but range shifts caused most size re-structuring of assemblages, due to uphill shifts of relatively small species, especially at high elevations.’

When one examines Figure 1 though, whether or not the authors intended this, one immediately comes away with the impression that bigger moths have moved into higher elevations in 2007. I have stared at this figure for a fair while and could not get my mind to change.

Thus, the authors might want to rethink this figure.

Author response [24]: We apologise for the confusion. The original figure 1 illustrates only one species and how it may contribute to the new size structure of assemblages along the elevation gradient, in relation to range shifts and intra-specific size changes. We agree that the different sizes of the moths can be mistaken as illustrating different species and so confuse readers. To facilitate interpretation, we have now moved it to figure 2 and place a stronger emphasis on how we consider range shifts and body size changes for each species in this study, particularly how we define each species’ movement (range shifts and non-boundary dynamics). The elevational size cline is now removed for clarity. Please see below for the new version:

Figure 2. Conceptual scheme of how we consider range shifts and body size changes for each species in this study. Each moth species can potentially alter the composition of new assemblages by range shifts (expansions or contractions at upper- or lower- boundaries) as well as non-boundary dynamics (local extinction, colonization or persistence). The size structure of the new assemblages will be determined by changes in species composition at sites and intra-specific body size changes of each species. As species are expected to reduce body size and move uphill under warming, assemblage size structures will change accordingly.

[25] Reviewer #2 (Remarks to the Author):

The authors have largely addressed the previous concerns I had with their paper, to where the manuscript is now clear in their presentation of an interesting and important set of ideas and data that will be of broad interest. Below are several mostly minor suggestions that I think might improve the manuscript.

Line 28: Having the word 'tropical' is awkward here - makes me start to wonder about latitudinal effects (which you don't explore at all). I'd remove, or move to another sentence.

Author response [25]: We have revised the abstract and "tropical" has been removed.

[26] Line 34: Are you referring to the 0.5% and 0.6% effects? It's not currently clear to me.

Author response [26]: The abstract has been revised.

[27] Line 39: I don't like the word 'individualistic' here, as certainly there are some general rules emerging (as your current paper shows). You could just replace it with 'variable'

Author response [27]: Revised accordingly.

[28] Line 44: Are you not also talking about changes to the size of species as well?

Author response [28]: We discuss size shrinkage of species in the following paragraph.

[29] Line 89-90: Is there are reference for this?

Author response [29]: We have added a reference (Chen et al. 2009).

[30] Line 98: In the caption for Fig. 1 you refer to three main processes - but you just list two here. It's confusing, so try to be consistent where possible.

Line 98-102: What the four categories of species composition changes are is unclear from the text to me - sentence needs reworking to make clear.

Line 104-105: Moving this to earlier in the paragraph - where you can start by stating that there are a total of 6 processes - might help clarify the issues I identify in the above couple of comments.

Line 109-111: Ditto to my previous comments. This just adds to the confusion. I might integrate above, or at minimum rework all this text to be clearer about how you are defining/categorizing processes (and how it links to Fig. 1)

Author response [30]: This paragraph has now been moved to Results to follow the article structure of Nature Communications, and has been revised to improve clarity. The original Fig. 1 has been edited and is now Fig.2. Line 109-111 has been removed. The paragraph now reads (L123-134):

“The size structures of assemblages are determined by species composition at sites and the body size of each species. Each moth species can potentially alter the composition of new assemblages by range shifts (expansions or contractions at

upper- or lower- boundaries) as well as non-boundary dynamics (local extinction, local colonization or local persistence). Thus, each species will contribute to the new assemblage size structure by range shifts and non-boundary dynamics, as well as by intraspecific size changes under warming (illustrated conceptually in Fig. 2). For each assemblage, in order to estimate how these processes shape the new size structure, we re-computed the 2007 assemblage size by allowing just one of the following processes to occur at a time (intraspecific size change, four categories of range shift, and non-boundary dynamics). We obtained means and errors for each process's contribution to community size structure change from 500 re-samplings of the data set to account for differences in sample effort between surveys (see methods for details)."

[31] Line 118: Can you point to some Figs/Tables in the supp info that support this statement?

Author response [31]: Please refer to our response [8] above.

[32] Line 124: I think refer to Fig. 2a at the end of this first sentence, and maybe remove it from the next sentence. I think do this because it is difficult to estimate the changes at each elevation from the data as presented in Fig. 2a.

Author response [32]: Revised accordingly.

[33] Line 128-129: I think delete 'where the average body sizes of assemblages were reduced by 0.79 mm to 2.13 mm 128 (-4.6% to -11.7%)' as its mostly repetitive from the previous sentence.

Author response [33]: We prefer to keep this sentence as it refers only to the highest three sites, which are our main focus.

[34] Line 133-136: While I like the idea of Fig. 2c, I'm not sure it's the best way to highlight the trend you describe here - it's very tough to tell this trend from the way the data is presented.

Author response [34]: We now cite this figure (now Fig. 1c) in the previous sentence when we first explain the variation we are describing:

"Coefficients of variation increased significantly at five of the seven sites, although

one site (at intermediate elevation, Radio Sabah, 2685m a.s.l.) showed a significant decrease (Fig. 1b and c, Supplementary Table 2)."

[35] Line 140: Personally, I'd like to see Fig. A1.1 in the main text

Author response [35]: This figure shows the frequency distribution of changes in forewing length for each species whereas the main conclusion of the analysis is that intraspecific size change (1.3% reduction) is not as important as range shifts. We thus prefer to keep this figure in the supplementary material to prevent confusion.

[36] Line 142: I'm confused, shouldn't this 4/9% be 0.6% (which i got from your abstract). If I am confused, then it's probably because of your inconsistency is how you refer to different processes/sets of processes through the manuscript.

Author response [36]: This paragraph has been revised and the sentence was deleted.

[37] Line 156-157: Delete "over the 42-year period along the 156 elevation transect."

Author response [37]: We have deleted this sentence.

[38] Line 164: Maybe also or instead say 'ecological interactions' (just to be broader than plant quality, as it could also be predators, or parasites, or competition, etc.)

Author response [38]: We have added "potential ecological interactions" to this sentence.

[39] Figure 1: This figure is a good addition, but I think it could be made clearer and more complete. What about a third column on the right that explains the effect (at each elevational band) of the process on assemblage size structure? It would also be great if you could add the three processes you mention in the caption to the figure (i.e. intraspecific size change, boundary shifts, and within range sizes).

Author response [39]: Many thank for the suggestions. This Figure has now been re-drawn (now fig. 2) to make the categories more clear. We decided to simplify it rather than adding more information. Please also refer to our response above [24].

[40] Reviewer #3 (Remarks to the Author):

The authors have done a reasonable job of addressing the issues raised. I have no further major comments.

Author response [40]: We thank all the reviewers for their constructive comments that have helped us to improve the manuscript.